computer modelling and simulation/ecology/statistical physics

robustness, extinctions, network models, evolutionary dynamics, dormancy

**Author for correspondence:**
Fumiko Ogushi
e-mail: ogushi.fumiko.54n@st.kyoto-u.ac.jp

# Temporal inactivation enhances robustness in an evolving system

Fumiko Ogushi[1,2], János Kertész[3,4], Kimmo Kaski[5,6] and Takashi Shimada[7,8]

[1]Kyoto University Institute for Advanced Study, Kyoto University, Yoshida Ushinomiya-cho, Sakyo-ku, Kyoto 606-8501, Japan
[2]Center for Materials Research by Information Integration, National Institute for Materials Science, 1-2-1 Sengen, Tsukuba, Ibaraki 305-0047, Japan
[3]Department of Network and Data Science, Central European University, 1051 Budapest, Hungary
[4]Institute of Physics, Budapest University of Technology and Economics, 1111 Budapest, Hungary
[5]Department of Computer Science, Aalto University School of Science, PO Box 15500, Espoo, Finland
[6]The Alan Turing Institute, British Library, 96 Euston Road, London NW1 2DB, UK
[7]Mathematics and Informatics Center, and [8]Department of Systems Innovation, Graduate School of Engineering, The University of Tokyo, 7-3-1 Hongo, Bunkyo-ku, Tokyo 113-8656, Japan

FO, 0000-0002-5055-4023

We study the robustness of an evolving system that is driven by successive inclusions of new elements or constituents with $m$ random interactions to older ones. Each constitutive element in the model stays either active or is temporarily inactivated depending upon the influence of the other active elements. If the time spent by an element in the inactivated state reaches $T_W$, it gets extinct. The phase diagram of this dynamic model as a function of $m$ and $T_W$ is investigated by numerical and analytical methods and as a result both growing (robust) as well as non-growing (volatile) phases are identified. It is also found that larger time limit $T_W$ enhances the system's robustness against the inclusion of new elements, mainly due to the system's increased ability to reject 'falling-together' type attacks. Our results suggest that the ability of an element to survive in an unfavourable situation for a while, either as a minority or in a dormant state, could improve the robustness of the entire system.

# 1. Introduction

The robustness of a system with many interacting elements or constituents under successive addition of new elements is an essential question for understanding the behaviour of various

complex real-world systems, that are often called ecosystems. (Here the term 'ecosystem' is used in a rather general sense to mean biological ecosystems but also diverse economical and social systems of individuals and institutions.) [1,2]. In these systems, the interactions between elements can be competitive or cooperative in nature such that the fitness of its elements or species can be strengthened or weakened by them, possibly causing the species to become extinct. This problem calls for a network theoretic approach, where the constituents of the system are the nodes of a dynamical network and the interactions are the links between them [3–6]. Then the rephrased question is about the evolution of such a network of nodes under the condition that new nodes with different kinds of links are introduced. If the network can grow, then the evolving system it describes is considered robust, otherwise the system does not grow and is considered volatile. This way, we believe that the network approach can be used and be versatile in investigating various aspects of robustness for a wide range of different systems.

Earlier it has been shown that in a simple model setting, where directed random positive and negative interactions characterize the system and the fitnesses of nodes (i.e. species) are identified with their strengths, when the links per node ratio—serving as a critical parameter—remains within a certain range, the system is robust [7]. This mechanism and the resulting phase diagram of the growth of the system were found to be universal, i.e. this feature is shared among a variety of models like the one with different distributions of interaction weights and with constant or random number of links introduced with the new nodes [8] and even with different bidirectional correlations [9]. While the range of robustness may be influenced by the details of the model, e.g. the mutuality in the interactions increasing it, the overall picture remains the same.

An alternative way to study the problem of robustness in complex interacting systems is a population dynamics-based approach as often done in theoretical ecology [10–12]. Such a framework enables more complex dynamics and is flexible with respect to allowing different states of the species, but unlike in the network approach, the inclusion of topological constraints are less straightforward in the population dynamics approach. Our aim here is to contribute to the convergence of these different approaches by including complex temporal features of interactions into the network models.

In population dynamics models, less fit species become minor in their population which in general makes that species almost irrelevant to the other species before that really gets extinct. For example, in the well adopted (generalized) Lotka–Volterra model [13,14] and replicator dynamics model [15], the trajectory starting from a feasible initial state (i.e. all population variables are positive [16]) never touches 0 within finite time. Therefore, a threshold is generally introduced to model extinction. This is a simplified treatment of the Allee effect [17] about the weakening of the fitness in small populations, or rather direct modelling of the negative effect of demographic stochasticity [18,19]. In summary, these observations and the related approaches suggest that the population size of less fit species and its temporal derivative becomes very small before extinction and the process is often lengthy. Furthermore, the adaptive nature of foraging and other interactions at the population level and at the individual level [12,20–24] make such very minor species effectively even more invisible for other species. Therefore, it seems plausible to include an 'inactive state' into the set of possible states for handling such weakened populations. Species in such an inactive state, i.e. close to extinction, could be revived or reactivated within a frame of time if the circumstances would sufficiently improve.

The introduction of an inactive state can be also regarded as modelling *dormancy*, which is broadly observed in biological ecosystems, such as in the case of hibernation and surviving in seed, spore or bacterial spore [25,26]. From the evolutionary point of view hibernation or dormancy is favourable as it enables survival under scarce conditions. Therefore, we expect that this new component if considered in the framework of network models will increase the robustness of the system, which in turn should be reflected in the increase of the growth region in the phase diagram.

The paper is organized such that in the next section, we describe our network-based model of evolutionary system of species capable of being temporarily inactive. This is followed with a comprehensive account and analysis of computational modelling results to map out the phase diagram of the evolutionary system. Then we draw conclusion and present discussions.

## 2. Model

As we consider the ecosystems as being composed of connected species, we have devised our model being a network of nodes (or species) connected by unidirectional links with weights, as illustrated schematically in figure 1. Here the nodes represent species of animals of some sort and the links

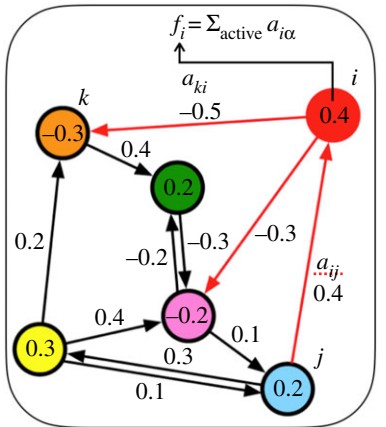
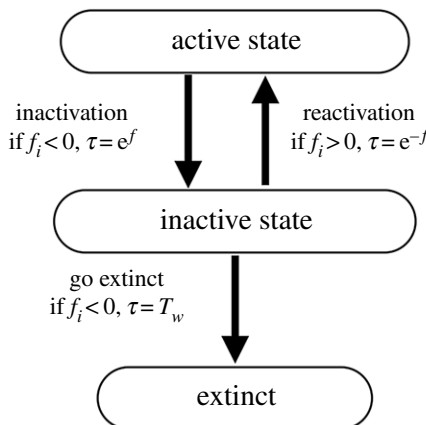

**Figure 1.** Introduction of the inactive state (dormancy) before the extinction, to our graph-dynamics framework. Less fit species is inactivated faster, and better fit species in inactive state is reactivated faster. The time limit of dormancy till extinction is, in contrast, uniformly set to $T_W$.

different types of directed influences between the pairs of species. The strength of the influence of species $j$ on species $i$ is denoted by the weight of the unidirectional link from node $j$ to node $i$, i.e. $a_{ij}$. These weights can be either positive or negative. Each species has its 'fitness', which is simply given by the sum of its incoming interactions from other species in the system, i.e. $f_i = \sum_j^{\text{incoming}} a_{ij}$. A species can survive as long as its fitness is greater than zero. The species with non-positive fitness, which in our previous models [7,9] went instantaneously extinct, will in the present model be *inactivated* after its fitness-dependent waiting time $\tau = e^f$, i.e. species in worse situation is inactivated faster. The inactivated species loses its influence on other species; thus, we will neglect the links out of those for the calculation of fitness. If the surrounding community of an inactivated species changes and the fitness of an inactivated species becomes positive, the species is reactivated (waking up from dormancy). The waiting time of this reactivation process is also assumed to be fitness-dependent: $\tau = e^{-f}$. The slowest process among the microscopic dynamics is the inactivation and reactivation of solitary species ($f = 0$). The duration of these processes, $\tau = 1$, gives the unit of time to this otherwise time-scale-less model. Although it is known that some species can maintain its dormancy for quite a long time [27], the period has generally a limit. In the following, we introduce a uniform time-limit parameter $T_W$. A species that has spent $T_W$ of continuous time in the inactive state with non-positive fitness gets extinct. The extinct species and its incoming and outgoing links are removed permanently. Note that the present model with dormancy reduces to the original model [7] at $T_W = 0$. A pseudo-code style description of the entire dynamics is available in appendix A.

An example of temporal evolution of the system is shown in figure 2. If all the species are in active state and have positive fitnesses, nothing will happen. Therefore, we call such a state a *persistent state*. In the previous models [7,9], we added a new species every time the community has reached a persistent state. This corresponds to a low-introduction (mutation, invasion, etc.) rate limit. In the present model, however, it is also possible that the system relaxes to a limit cycle and never reaches a stationary persistent state (figure 3). Therefore, we need a new parameter for the time interval of the species introduction, $T_{\text{int}}$. In the following, we take a long interval: $T_{\text{int}} = 100$ to keep a low-introduction rate, unless otherwise noted.

Every time after finding a persistent state or elapsed time $T_{\text{int}}$, we proceed to the next time step by adding a new species with $m$ interactions into the system. The $m$ interacting species are chosen at random from the resident species with equal probability and the directions (incoming or outgoing) are also determined at random. The link weights are again assigned at random from the standard normal distribution.

## 3. Results

Following the approach of our previous study [7], we assess the robustness of the emergent system by the long-term trend of the system size, i.e. the number of species, under the successive introduction of new

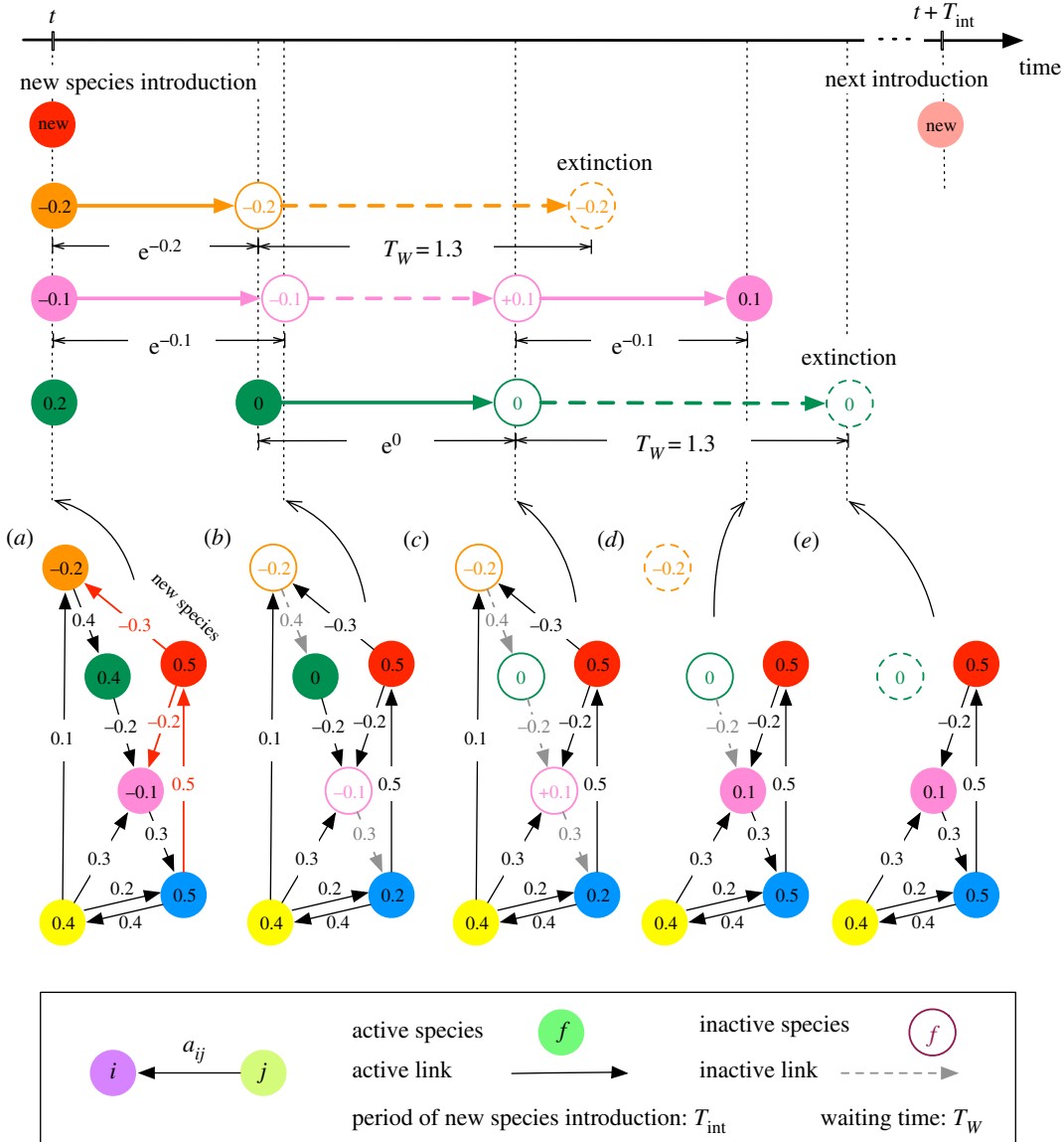

**Figure 2.** A temporal evolution of the model with inactivation (dormancy) and reactivation (revival), after inclusion of new species. (*a*) Introduction of a new species (red), which makes the fitness of two species (orange and magenta) negative. Each of these two species will be inactivated after its fitness-dependent duration: $\tau = e^{f_i}$. (*b*) Inactivation of the species with worse fitness (orange) takes place first and then the other species (magenta) is inactivated, which makes the fitness of another species (green) non-positive. Inactivated species is given $T_W$ of waiting time till it will go extinct. (*c*) Green species is inactivated before any of other inactive species goes extinct. This change makes the fitness of the inactive species (magenta) positive. (*d*) Magenta species is reactivated after a fitness-dependent waiting time $\tau = e^{f_i}$. Meanwhile, the orange species have spent $T_W$ of time in the inactivated state and hence gone extinct: the orange species and the interactions from and to it are deleted. (*e*) Green species goes extinct. This does not change the sign of fitness of any species in the community. Therefore, after the extinction of green species, the system finally reaches a new persistent state, i.e. all the species are in the active state and have positive fitnesses. Nothing will happen for a community in a persistent state, until the next new species is introduced at $t + T_{int}$.

species which, in terms of the directions and the weights of its interactions, has neutral effect on growth. In our original model without any dormant mechanism [7], the system can grow limitlessly; thus, it is robust enough against the inclusion of new species if the number of interactions given for each newly introduced species, $m$, is kept within a moderate range, i.e. $5 \leq m \leq 18$. By contrast, the system with $m$ outside this range, keeps fluctuating with a finite size. These fluctuations may lead to the extinction of the entire system, and the lower the mean level is, the higher is the probability for such an event. To avoid this possibility, we adopt an incubation rule when the system size becomes smaller than the initial system size $N_0$. Under the incubation rule, we let totally isolated species (i.e. $f_i = 0$) stay in

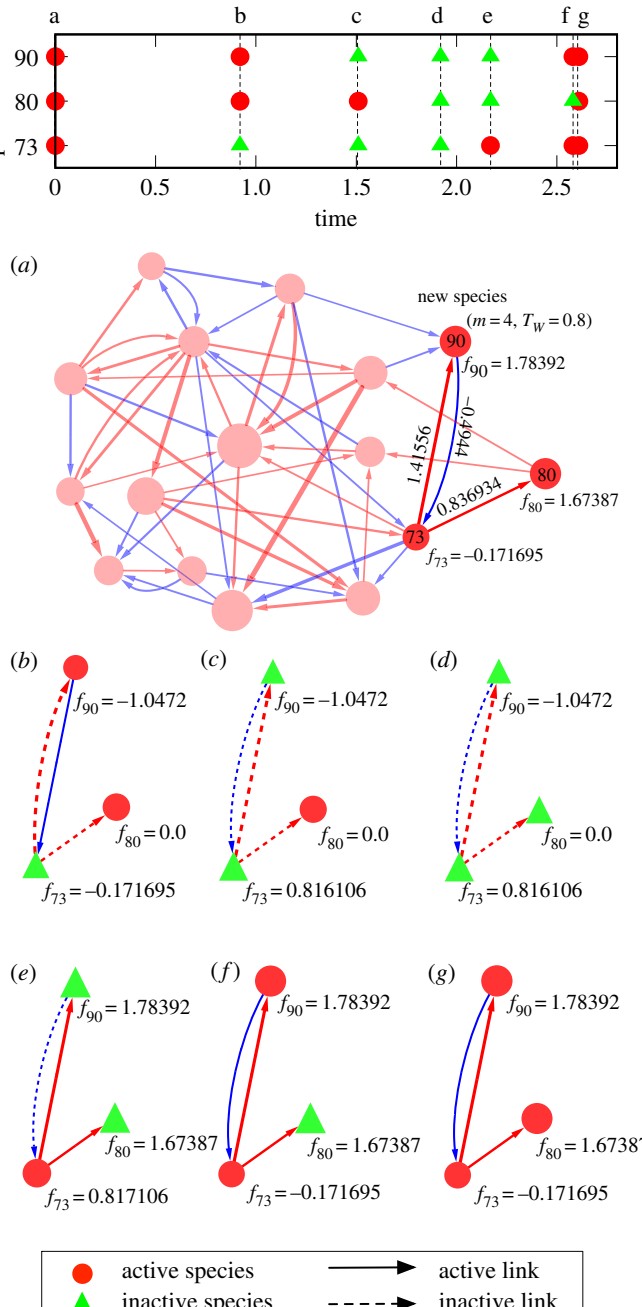

**Figure 3.** A limit cycle observed in an emergent system in the present model with inactivation and revival processes. (Top) The time series of the state changes of the species involved in the limit cycle (73, 80 and 90). (a) The entire system at the beginning of the cycle, $t = 0$. Red and blue arrows represent positive and negative interactions, respectively, such that the thickness represents the amplitude of the interaction. Precise values of the interactions are shown only for the interactions among the species 73, 80 and 90. (b) Species with negative fitness, 73, is inactivated at $t = \exp\{f_{73}\}$. This makes the fitness of species 80 and 90 non-positive. (c) Species with worse fitness, 90, is inactivated. This makes the fitness of species 73 positive. (d) Species 80 is later inactivated. (e) Species 73 is reactivated before the extinctions of species 80 and 90, making the fitness of those species positive. (f) Species 90 is reactivated. (g) Species 80 is reactivated and those three species come back to the initial all-active state.

the active state or inactive state. This treatment prevents the total collapse of the system and provides the system with many more opportunities to search for growth from different initial conditions.

For sufficiently large initial system size, typically $N_0 \geq 100$, the limitless growth and finite size fluctuation behaviour are confirmed to be independent of the initial network structure. Therefore, we call the former behaviour taking place in the 'diverging phase' and the latter in the 'finite phase' of the parameter space. The temporal evolution of the system size of the present model with $m = 25$ is

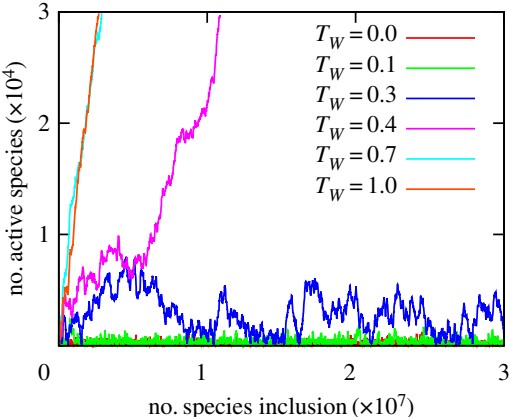

**Figure 4.** The temporal evolutions of total number of active species $N_{active}(t)$ under the successive introduction of new species with $m = 25$ interactions. The unit for time is $T_{int}$, i.e. the horizontal axis corresponds to the accumulated number of introduced species. The size of the emergent system diverges in time if the waiting time of dormancy is long ($T_W \geq 0.4$) while it fluctuates within a finite size for shorter waiting time ($T_W \leq 0.3$).

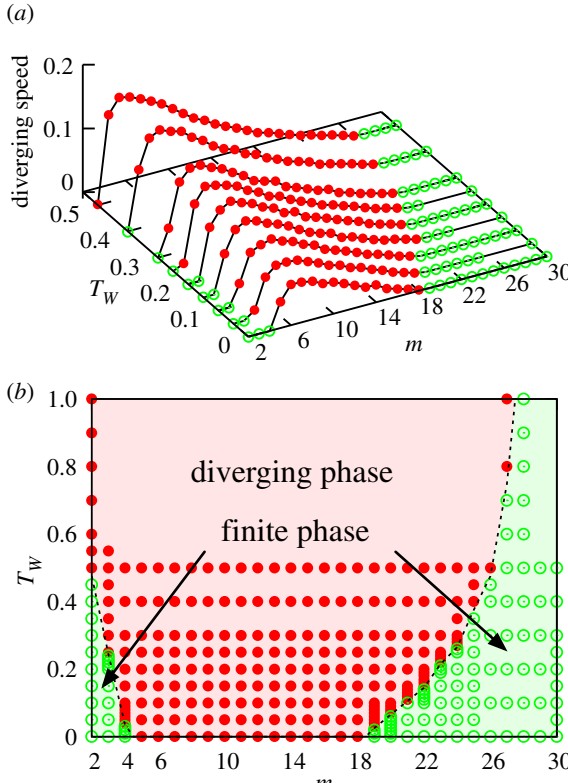

**Figure 5.** The phase diagram of the evolving open system with dormancy and revival processes. (a) The speed of divergence $v = \lim_{t \to \infty} N(t)/t$ for the given original and new key parameters, $m$ and $T_W$. The points where $v$ is evaluated to be positive are shown by filled red symbols. (b) The corresponding phase diagram.

shown in figure 4. Inheriting the nature of the original model [7], the system with short dormancy limit $T_W$ is found to be in the finite phase. However, as $T_W$ increases (to the value $T_W = 0.3$) the typical system size shows a clear increase yet it stays finite, and for $T_W = 0.4$ and above the system has crossed a certain threshold to show diverging behaviour. This clearly illustrates that our newly introduced parameter $T_W$, the time limit for the continuous dormancy, can change the robustness of the system.

Next we will explore the whole phase diagram with systematic computer simulations by scanning through the $m$ versus $T_W$ parameter space. The obtained phase diagram is shown in figure 5, where it is seen that the introduction of dormancy and revival processes broaden the diverging phase. While

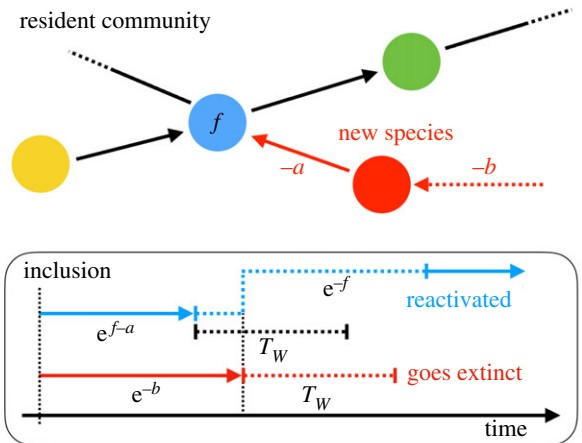

**Figure 6.** The mechanism of rejecting the attack by species with non-positive fitness.

this effect turns out to be larger for longer dormancy time limit $T_W$, yet it is not possible to get the system with very dense interactions ($m \geq 28$) to the diverging phase. It should also be noted that, as shown in appendix A, the basic network characteristics of the emergent systems are not so much dependent on $T_W$ and not deviated from that of Erdős–Rényi random graph, indicating the almost random network structure as in the previous models [7,9].

The main mechanism of this enforcement is the rejection of 'falling-together-attacks'. To illustrate this, let us consider a situation that a negative link weight ($-a$) is added to a resident species by a newly introduced species, which has zero or negative fitness value, $-b$ (figure 6).

In the original model [7] and in the present model with $T_W = 0$, in which the least fit species goes extinct first, the attacked resident species and the new species sequentially go extinct for $f - a < -b$ and otherwise only the new species goes extinct (i.e. is rejected). Especially for the newly introduced species with no incoming links ($b = 0$, solitary attack), every attack strong enough ($f < a$) can kill the resident species before the newly introduced attacker species goes extinct.

In the present model with $T_W > 0$, the situation is different as the resident species has another chance to reject such a falling-together attack. The rejection happens if the resident species can survive in the inactivated state until the newly added species stays inactivated. The condition for this type of dynamics is as follows:

$$f - a < -b < \ln(e^{f-a} + T_W). \tag{3.1}$$

Therefore, even a strong attack ($f > a$) by a solitary new species ($b = 0$) is rejected if $T_W > 1 - e^{f-a}$. And if $T_W \geq 1$, i.e. the limit of the dormancy period is long enough, even the solitary attacks never become successful. Note that the rejection acts perfectly in a special case of $m = 1$, because in this situation every inclusion of new species corresponds to either a solitary attack or an attachment of species with no outgoing link. Therefore, even for this most sparse condition, large $T_W$ drives the system with a mutually supporting community core to grow infinitely in size. However, such a growth is highly dependent on the initial condition (if there is no core in the initial network, the system collapses) which is out of the scope of this study. Thus, we excluded this case from the phase diagram.

The increment of probability to reject falling-together-attacks directly contributes to the growth rate of the system, $v = N(t)/t$. A rough estimate of it near the upper phase boundary ($m \sim 18$) predicts a linear increase of the rejections to $T_W$ for the small $T_W$ regime (see appendix A for details), which is confirmed in the simulation (figure 7). The observed contribution of the additional rejections to the system's growth rate, $\Delta v \sim T_W/8$, predicts the slope of the phase boundary to behave as $\Delta m^* \sim 20\, T_W$. This is found to be consistent with the phase portrait.

The effect of rejections in the sparse regime ($m \leq 4$) needs to be estimated differently. This is because the probability to have a solitary attack is larger. What is more significant, however, is the fact that the resident community has a sparse network structure, which in turn is very prone to a loss of certain species and can cause a cascade of extinctions of species supported by that species. Therefore, the effect of the increased chance of rejection can be more drastic. It is also possible that the structure of the emergent networks is changed, although the basic network characteristics (see appendix A) and the well-kept distributions of extinction cascade size suggests it to be negligible at least for $m = 4$

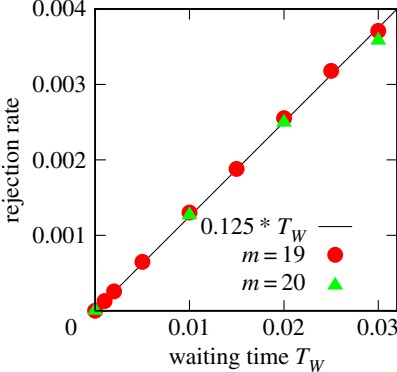

**Figure 7.** The rejection rate obtained from the simulation in the dense regime. In the small $T_W$ regime shown here, the rejection rate increases linearly to $T_W$.

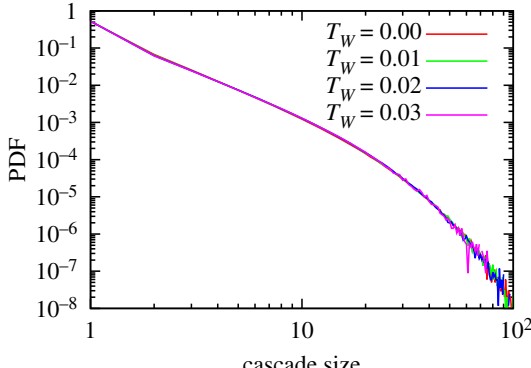

**Figure 8.** The cascade size distributions of the extinctions in the model with $m = 4$. The distributions from the systems in the finite phase ($T_W \leq 0.02$) and from the diverging phase ($T_W = 0.03$) overlap well each other, indicating the structure of the emerging networks is kept.

(figure 8). The consideration above predicts the broadening of the diverging phase, but it is difficult to give an estimate of the effect of $T_W$ against the very steep drop of the growth rate in this regime of the phase diagram.

## 4. Summary and discussion

We have studied the robustness of an evolving system against successive inclusions of new elements or constituents, each with an ability to survive temporarily under unfavourable conditions in the state of being inactive. It is found that the introduction of the inactivation and revival processes broadens the phase the systems stays robust. This reinforcement of the emerging system is mainly due to its increased ability to reject falling-together type attacks. It should be noted that the broadening of the robust phase has a limit: systems with $m \geq 28$ stay in the finite phase even at $T_W = 1$, where the rejection probability reaches its maximum. The short-term rejection process, in which a possible extinction of a species caused by the attack from a species with poor fitness is altered by the extinction of the attacker, can be regarded as a simplified dynamics in a class of population dynamics models [13–15,22]. Because another type of interaction form, namely the ratio-dependent interaction [28], is known to reduce to our previous model [29], the extension of the model in this study has broadened the applicability of our theoretical framework. Similar to our earlier results [7,9], we have found that the number of interactions per species limits the system's robustness. There are empirical findings in support to this observation [30].

As for the modelling in general the population dynamics models based on differential or difference state equations are able to describe rich evolutionary patterns following periodic and even chaotic

trajectories, as observed in nature [31,32]. However, this approach is generally computationally so costly that larger system sizes and longer time scales could not be studied. In order to circumvent these problems, we have taken a network-based approach, which is able to describe the dynamics of the system over much longer evolutionary time scale.

Although our present analysis covers up to the long-dormancy time limit ($T_W = 1$) in terms of the resulting short-term rejection process, far longer dormancy limit ($T_W \gg T_{int}$) could bring new phenomena. Under such condition, inactive species can survive evolutionary time scale during which new species are introduced and that change the community. In some cases and for various kinds of systems, such as biological, social and economic systems, it may be important to consider such long dormancy periods [33]. Also, the effect of bidirectionality [9] of the interaction should be examined, because it is expected to make the emergent system show limit cycles more frequently. These two regimes, although that requires heavier computation power, will reveal new phenomena and will better bridge with the continuous time dynamics models. Extending our approach so that some aspects of short-term dynamics of more complex models are kept, with further spacial extension focusing on some aspects hardly accessible by traditional methods, is a promising way to treat evolutionary problems better [34,35].

Data accessibility. The simulation code for our model has been uploaded as the electronic supplementary material.

Authors' contributions. F.O. and T.S. conceived the model and conducted the simulation. All authors analysed the results and wrote the manuscript.

Competing interests. We declare we have no competing interests.

Funding. F.O. was partly supported by 'Materials Research by Information Integration' Initiative (MI2I) project of the Support Program for Starting Up Innovation Hub from the Japan Science and Technology Agency (JST). K.K. acknowledges financial support by the Academy of Finland Research project (COSDYN) no. 276439, EU HORIZON 2020 FET Open RIA project (IBSEN) no. 662725, EU HORIZON 2020 INFRAIA-1-2014-2015 program project (SoBigData) no. 654024 and the Rutherford Foundation Visiting Fellowship at The Alan Turing Institute, UK. T.S. was partly supported by JSPS KAKENHI grant nos. 15K05202 and 18K03449.

Acknowledgements. F.O., J.K. and T.S. thank for hospitality of Aalto University.

Disclaimer. Any opinions, findings or conclusions are those of authors.

# Appendix A

## A.1. Model procedure

(0) (Create an initial system)

 (i) Prepare $N_0$ species and connect them randomly by $L_0$ unidirectional links with link weights denoted by $a_{ij}$. Typical settings are $N_0 = 100$ and $L_0 = 10\,N_0$.

 (ii) Each species has its state variable ($S_i = \{-1, 1\}$, 1 and $-1$ denote active and inactive states, respectively), the time counters for state change $g_i$, and the counter for extinction $h_i$. Those are set to the initial values: $\{S_i\} = 1$, $\{g_i\} = 1$, $\{h_i\} = T_W$.

 (iii) Set the system time at $t = 0$ and the time for the next new species introduction $T_{next} = T_{int}$.

(1) Calculate the fitness $f_i$ of each species,

$$f_i = \sum_{j}^{\text{incoming}} \left( \frac{1 + S_j}{2} \right) a_{ij}.$$

(2) Reset the time counter if needed:

$$\begin{cases} g_i = 1 & (S_i = +1 \text{ and } f_i > 0 \cap f_i^{old} \le 0) \\ g_i = 1 & (S_i = +1 \text{ and } f_i \le 0 \cap f_i^{old} > 0) \\ g_i = 1 & (S_i = -1 \text{ and } f_i > 0 \cap f_i^{old} \le 0) \\ h_i = T_W & (S_i = -1 \text{ and } f_i \le 0 \cap f_i^{old} > 0), \end{cases}$$

where $f_i^{old}$ is the fitness at the previous time step.

(3) Calculate the remaining time till the next event for each species, $\delta t_i$:

$$\delta t_i = \begin{cases} T_{next} - t & (S_i = +1, f_i > 0: \text{ no state change}) \\ g_i e^{S_i f_i} & (S_i = +1, f_i \le 0: \text{ inactivation}) \\ g_i e^{S_i f_i} & (S_i = -1, f_i > 0: \text{ reactivation}) \\ T_{next} - t & (S_i = -1, f_i \le 0: \text{ extinction}). \end{cases}$$

(4) Find the shortest time to the next event in the system: $\delta t_j^* = \min\{\delta t_i\}$.

(5) Time translation of the system from $t$ to $t + \delta t_j^*$:

 (i) Update the system time $t = t + \delta t_j^*$;

 (ii) Update the time counters:

$$
\begin{cases}
g_i = g_i & (S_i = +1 \text{ and } f_i > 0) \\
g_i = g_i - e^{S_i f_i} \delta t_j^* & (S_i = +1 \text{ and } f_i \leq 0) \\
g_i = g_i - e^{S_i f_i} \delta t_j^* & (S_i = -1 \text{ and } f_i > 0) \\
h_i = h_i - \delta t_j^* & (S_i = -1 \text{ and } f_i \leq 0)
\end{cases}
$$

 (iii) Extinction: If $h_i \leq 0$, delete the species $i$ and all links connecting to and from it.

(6) Treat the event at $t$ (state change of species $j$ or new species introduction)

 — If $t < T_{\text{next}}$, treat the nearest state change of species, $j$:

 (i) Update the state of the species $j$:

 $S_j = -S_j$.

 (ii) Reset the time counters:

 $g_j = 1$ and $h_j = T_W$.

 — If $t = T_{\text{next}}$, add a new species:

 (i) The new species is added in active state ($S = +1$) with the time counters $g = 1$ and $h = T_W$.

 (ii) $m$ interacting species are randomly chosen from the resident species.

 (iii) The new species forms $m$ directed unidirectional links. The direction of each new link is chosen with a equal probability $1/2$.

 (iv) The link weights are also randomly chosen from a standard normal distribution.

 (v) Update the time for the next species introduction: $T_{\text{next}} = T_{\text{next}} + T_{\text{int}}$.

(7) Recalculate the fitness: go back to step (1).

## A.2. Estimation of the rate of the additional rejections and its effect

Here we first roughly estimate the increment of the chance to reject such falling-together-attack which directly contributes to the growth rate of the system, $v = N(t)/t$, near the upper phase boundary ($m \sim 18$). In the vicinity of the phase boundary in the dense regime, an inclusion of new species causes one strong attack ($f < a$) event in average. The distribution of $f - a$ is given by the negative side of the convolution:

$$
\rho(f - a) = \int_0^\infty \bar{f}(\xi) \, G(1, f - a - \xi) \, d\xi, \tag{A 1}
$$

where $\bar{f}(x)$ and $G(\sigma, x)$ represent the equilibrium fitness distribution of the emergent system and the Gaussian distribution with its standard deviation $\sigma$, respectively. The distribution of the fitness of newly added species, $-b$, is well approximated by the negative half side of the Gaussian distribution $G\left(\sqrt{m/2}, -b\right)$, where $m/2$ is the average number of incoming links. For small $T_W$, the condition to have the dormancy-aided rejection, equation (3.1), is

$$
f - a < -b < f - a + \frac{T_W}{e^{f-a}}. \tag{A 2}
$$

Substituting $\rho(-b)$ near 0 by its peak value $G\left(\sqrt{m/2}, 0\right) = 1/\sqrt{\pi m}$, and taking $f - a \sim -1/2$ as a typical attack strength, an estimated increment in the system's growth rate brought by the increase of the rejection is

$$
\Delta v_{\text{est.}} \sim \sqrt{\frac{e}{\pi m}} T_W \sim \frac{T_W}{5}. \tag{A 3}
$$

We can confirm this linear relationship between the rejection rate $\Delta v$ and $T_W$ in the simulation results for

$m = 19$ and 20 (figure 7). And the observed slope

$$\Delta v_{\text{obs.}} = \frac{T_W}{8} \tag{A 4}$$

is also consistent with the very rough estimation above.

Taking the linear slope of the system's intrinsic growth rate to $m$ obtained from the observed growth rates

$$\frac{\Delta v}{\Delta m} \sim \frac{0.06}{10}, \tag{A 5}$$

we reach an estimation for the slope of phase boundary

$$\Delta m \sim 20\, T_W. \tag{A 6}$$

## A.3. Network characteristics of the emergent systems

The network characteristics of emergent systems are shown in table 1.

**Table 1.** Network characteristics of emergent systems. For each parameter set, $(m, T_W)$, the average and the error are calculated from 10 independent simulation samples. Samples in the diverging phase are taken when the system size reaches $N_{\text{obs}} = 20\,000$.

| $m$ | $T_W$ | $\langle N \rangle$ | $\langle k \rangle$ | assortativity | nestedness[a] | clustering coefficient[a] |
|---|---|---|---|---|---|---|
| 19 | 0.5 | $N_{\text{obs}} = 20000$ | $20.11 \pm 0.05$ | $-0.024 \pm 0.001$ | $1.0909 \pm 0.0005$ | $0.919 \pm 0.005$ |
| | 0.3 | $N_{\text{obs}}$ | $20.04 \pm 0.05$ | $-0.0240 \pm 0.0007$ | $1.0884 \pm 0.0003$ | $0.928 \pm 0.009$ |
| | 0.1 | $N_{\text{obs}}$ | $19.68 \pm 0.05$ | $-0.022 \pm 0.001$ | $1.0874 \pm 0.0004$ | $0.912 \pm 0.007$ |
| | 0.0 | $2.3 \times 10^3$ | $19.36 \pm 0.01$ | $-0.0197 \pm 0.0001$ | $1.0873 \pm 0.0001$ | $0.9172 \pm 0.0001$ |
| 10 | 0.5 | $N_{\text{obs}}$ | $11.55 \pm 0.03$ | $-0.028 \pm 0.001$ | $1.1312 \pm 0.0009$ | $0.87 \pm 0.02$ |
| | 0.3 | $N_{\text{obs}}$ | $11.39 \pm 0.02$ | $-0.030 \pm 0.001$ | $1.1288 \pm 0.0007$ | $0.89 \pm 0.02$ |
| | 0.1 | $N_{\text{obs}}$ | $11.10 \pm 0.02$ | $-0.0338 \pm 0.0007$ | $1.1205 \pm 0.0007$ | $0.83 \pm 0.02$ |
| | 0.0 | $N_{\text{obs}}$ | $10.89 \pm 0.03$ | $-0.036 \pm 0.001$ | $1.1188 \pm 0.0005$ | $0.86 \pm 0.02$ |
| 4 | 0.5 | $N_{\text{obs}}$ | $5.20 \pm 0.02$ | $-0.040 \pm 0.002$ | $1.204 \pm 0.002$ | $0.75 \pm 0.04$ |
| | 0.3 | $N_{\text{obs}}$ | $4.98 \pm 0.01$ | $-0.052 \pm 0.001$ | $1.186 \pm 0.001$ | $0.71 \pm 0.09$ |
| | 0.1 | $N_{\text{obs}}$ | $4.69 \pm 0.01$ | $-0.073 \pm 0.002$ | $1.160 \pm 0.001$ | $0.72 \pm 0.07$ |
| | 0.0 | $1.0 \times 10^3$ | $4.63 \pm 0.01$ | $-0.0746 \pm 0.0002$ | $1.1567 \pm 0.0003$ | $0.662 \pm 0.002$ |
| 3 | 0.5 | $N_{\text{obs}}$ | $4.04 \pm 0.01$ | $-0.058 \pm 0.002$ | $1.221 \pm 0.002$ | $0.59 \pm 0.09$ |
| | 0.3 | $N_{\text{obs}}$ | $3.79 \pm 0.01$ | $-0.081 \pm 0.001$ | $1.187 \pm 0.002$ | $0.65 \pm 0.05$ |
| | 0.1 | $5.6 \times 10^2$ | $3.78 \pm 0.01$ | $-0.0748 \pm 0.0009$ | $1.1954 \pm 0.0008$ | $0.608 \pm 0.006$ |
| | 0.0 | $4.3 \times 10^2$ | $3.85 \pm 0.01$ | $-0.063 \pm 0.002$ | $1.204 \pm 0.002$ | $0.65 \pm 0.01$ |
| 2 | 0.5 | $N_{\text{obs}}$ | $2.90 \pm 0.01$ | $-0.075 \pm 0.003$ | $1.256 \pm 0.002$ | $0.5 \pm 0.1$ |
| | 0.3 | $1.3 \times 10^3$ | $2.90 \pm 0.01$ | $-0.056 \pm 0.001$ | $1.265 \pm 0.001$ | $0.49 \pm 0.02$ |
| | 0.1 | $5.5 \times 10^2$ | $2.92 \pm 0.01$ | $-0.039 \pm 0.004$ | $1.271 \pm 0.005$ | $0.51 \pm 0.02$ |
| | 0.0 | $4.0 \times 10^2$ | $2.93 \pm 0.03$ | $-0.035 \pm 0.008$ | $1.296 \pm 0.007$ | $0.44 \pm 0.06$ |

[a]The clustering coefficient and nestedness are given in those ratio to the ones of Erdős–Rényi random graph with the same size and average degree.

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
