## [Reviewer comments · Royal Society Open Science]

Review History

RSOS-181471.R0 (Original submission)

Review form: Reviewer 1 (Reiji Suzuki)

Is the manuscript scientifically sound in its present form?

Yes

Are the interpretations and conclusions justified by the results?

Yes

Is the language acceptable?

Yes

Is it clear how to access all supporting data?

Yes

Do you have any ethical concerns with this paper?

No

Have you any concerns about statistical analyses in this paper?

No

Recommendation?

Accept with minor revision (please list in comments)

Comments to the Author(s)

This paper proposed a novel network model of ecosystems in which active and inactive (or dormant) state of nodes (species) can coexist. The model is well written, and the results clearly showed that the introduction of inactive states of species can significantly increase the robustness or growing process of the whole ecosystem, which is interesting.

One thing I would like to ask the authors is to discuss effects of T_w on the properties of network structures (e.g., average degree, degree distribution, average path length, clustering coefficient), because such analysis is basic and important in network sciences but there is almost nothing in the current manuscript.

In addition, there are some trivial comments for the improvement of the readability of the manuscript as follows:

-The authors mentioned their previous model (e.g., page 2, line 34, right), but it appears not always clearly explained in the manuscript. Could you clarify the original paper of that model and explain it briefly?

-The detail of FIG3 is not explained. Could you explain it in the main text or in its caption?

Review form: Reviewer 2 (Masanori Takano)

Is the manuscript scientifically sound in its present form?

Yes

Are the interpretations and conclusions justified by the results?

Yes

Is the language acceptable?

Yes

Is it clear how to access all supporting data?

Yes

Do you have any ethical concerns with this paper?

No

Have you any concerns about statistical analyses in this paper?

No

Recommendation?

Accept with minor revision (please list in comments)

Comments to the Author(s)

The authors aim to construct a model for studying the ecosystem of robustness with a small computational complexity.

For this purpose, they introduced an inactive state of species with low fitness to the models in their previous studies.

Inactive state is a feature in other ecosystem models which are computationally so costly.

The authors claim that their extended model with a small computational complexity showed qualitatively similar behavior to these high-cost models.

Development of low-cost models is important for complex systems which require large-scale simulations.

My evaluation is that the paper is publishable with minor scientific revisions.

My comments are as follows.

1. Introduction - 1st paragraph:

The authors described their questions.

They should provide some references for the questions because these questions are general problems in statistical physics and complex system science.

2. Model:

The authors extended their "original model".

However, they do not show a paper which introduced this model.

Which are papers [2-4] of the "original model"?

The authors should show the original paper and the differences between the original model and the extended model.

3. Model:

In the model, the parameter "a" showing relationships between species is important parameters.

However, the description of "a" is missing.

Please add description how the model provides each "a" is not described in the main manuscript.

4. Results:

The authors sometimes mention their previous studies.

However, each mention is unclear which paper it refers to.

For example, "our previous study" (L. 1 in 1st Paragraph) and "our original model without ..." (L. 4 in 1st Paragraph).

Here are the minor comments.

a. the footnote 1 and the reference 1 are duplicated.

==

Masanori Takano
CyberAgent, Inc.

Decision letter (RSOS-181471.R0)

04-Jan-2019

Dear Dr Ogushi

On behalf of the Editors, I am pleased to inform you that your Manuscript RSOS-181471 entitled "Temporal inactivation enhances robustness in an evolving system" has been accepted for publication in Royal Society Open Science subject to minor revision in accordance with the referee suggestions. Please find the referees' comments at the end of this email.

The reviewers and handling editors have recommended publication, but also suggest some minor revisions to your manuscript. Therefore, I invite you to respond to the comments and revise your manuscript.

- Ethics statement

- Data accessibility

<http://datadryad.org/submit?journalID=RSOS&manu=RSOS-181471>

- Competing interests

- Authors' contributions

AB carried out the molecular lab work, participated in data analysis, carried out sequence alignments, participated in the design of the study and drafted the manuscript; CD carried out the statistical analyses; EF collected field data; GH conceived of the study, designed the study,

coordinated the study and helped draft the manuscript. All authors gave final approval for publication.

- Acknowledgements

- Funding statement

Because the schedule for publication is very tight, it is a condition of publication that you submit the revised version of your manuscript before 13-Jan-2019. Please note that the revision deadline will expire at 00.00am on this date. If you do not think you will be able to meet this date please let me know immediately.

- 1) A text file of the manuscript (tex, txt, rtf, docx or doc), references, tables (including captions) and figure captions. Do not upload a PDF as your "Main Document";
- 2) A separate electronic file of each figure (EPS or print-quality PDF preferred (either format should be produced directly from original creation package), or original software format);
- 3) Included a 100 word media summary of your paper when requested at submission. Please ensure you have entered correct contact details (email, institution and telephone) in your user account;
- 4) Included the raw data to support the claims made in your paper. You can either include your data as electronic supplementary material or upload to a repository and include the relevant doi within your manuscript. Make sure it is clear in your data accessibility statement how the data can be accessed;
- 5) All supplementary materials accompanying an accepted article will be treated as in their final form. Note that the Royal Society will neither edit nor typeset supplementary material and it will

be hosted as provided. Please ensure that the supplementary material includes the paper details where possible (authors, article title, journal name).

on behalf of Dr Matjaz Perc (Associate Editor) and Miles Padgett (Subject Editor)
openscience@royalsociety.org

Reviewer comments to Author:
Reviewer: 1

Comments to the Author(s)

This paper proposed a novel network model of ecosystems in which active and inactive (or dormant) state of nodes (species) can coexist. The model is well written, and the results clearly showed that the introduction of inactive states of species can significantly increase the robustness or growing process of the whole ecosystem, which is interesting.

One thing I would like to ask the authors is to discuss effects of T_w on the properties of network structures (e.g., average degree, degree distribution, average path length, clustering coefficient), because such analysis is basic and important in network sciences but there is almost nothing in the current manuscript.

In addition, there are some trivial comments for the improvement of the readability of the manuscript as follows:

- The authors mentioned their previous model (e.g., page 2, line 34, right), but it appears not always clearly explained in the manuscript. Could you clarify the original paper of that model and explain it briefly?
- The detail of FIG3 is not explained. Could you explain it in the main text or in its caption?

Reviewer: 2

Comments to the Author(s)

The authors aim to construct a model for studying the ecosystem of robustness with a small computational complexity.

For this purpose, they introduced an inactive state of species with low fitness to the models in their previous studies.

Inactive state is a feature in other ecosystem models which are computationally so costly. The authors claim that their extended model with a small computational complexity showed qualitatively similar behavior to these high-cost models.

Development of low-cost models is important for complex systems which require large-scale simulations.

My evaluation is that the paper is publishable with minor scientific revisions.

My comments are as follows.

1. Introduction - 1st paragraph:

The authors described their questions.

They should provide some references for the questions because these questions are general problems in statistical physics and complex system science.

2. Model:

The authors extended their "original model".

However, they do not show a paper which introduced this model.

Which are papers [2-4] of the "original model"?

The authors should show the original paper and the differences between the original model and the extended model.

3. Model:

In the model, the parameter "a" showing relationships between species is important parameters.

However, the description of "a" is missing.

Please add description how the model provides each "a" is not described in the main manuscript.

4. Results:

The authors sometimes mention their previous studies.

However, each mention is unclear which paper it refers to.

For example, "our previous study" (L. 1 in 1st Paragraph) and "our original model without ..." (L. 4 in 1st Paragraph).

Here are the minor comments.

a. the footnote 1 and the reference 1 are duplicated.

==

Masanori Takano
CyberAgent, Inc.

Author's Response to Decision Letter for (RSOS-181471.R0)

See Appendices A - C.

Decision letter (RSOS-181471.R1)

15-Jan-2019

Dear Dr Ogushi,

I am pleased to inform you that your manuscript entitled "Temporal inactivation enhances robustness in an evolving system" is now accepted for publication in Royal Society Open Science.

on behalf of Dr Matjaz Perc (Associate Editor) and Miles Padgett (Subject Editor)
openscience@royalsociety.org

Appendix A

Response to reviewer 1:

We would like to thank the referee for the review of our manuscript and the points of improvements. We are very grateful for your constructive comments, which are helpful in improving the manuscript. We have made revisions based on all these comments and suggestions. The followings are our point-by-point responses.

(Point-by-point response to the referee)

Reviewer: 1

Reviewer's comment 1: This paper proposed a novel network model of ecosystems in which active and inactive (or dormant) state of nodes (species) can coexist. The model is well written, and the results clearly showed that the introduction of inactive states of species can significantly increase the robustness or growing process of the whole ecosystem, which is interesting.

Author's response: Thank you for finding our results clear and interesting.

Reviewer's comment 2: One thing I would like to ask the authors is to discuss effects of T_w on the properties of network structures (e.g., average degree, degree distribution, average path length, clustering coefficient), because such analysis is basic and important in network sciences but there is almost nothing in the current manuscript.

Author's response: Thank you for the comment. We have omitted the basic network statistics because it did not show any difference from what we had reported in the previous papers. But we agree that giving such information is important so we have added brief explanations in Result and a Table in the Appendix to show those information.

(the bottom of the third paragraph in Result, colored in red in the text)

... *It should also be noted that, as shown in Appendix, the basic network characteristics of the emergent systems is not so much dependent on T_w and not deviated from that of Erdős–Rényi random graph, indicating the almost random network structure as in the previous models [8, 10].*

(the bottom of the eighth paragraph in Result, colored in red in the text)

..., *although the basic network characteristics (see Appendix) and the well kept distributions of extinction cascade size suggests it to be negligible at least for $m = 4$ (FIG. 8).*

(Appendix C, colored in red in the text)

Network characteristics of the emergent systems are shown in Table 1.

(Table 1 and its caption in Appendix C, colored in red in the text)

Network characteristics of the emergent systems. For each parameter set, (m, T_w) , the average and the error are calculated from 10 independent simulation samples. Samples in the diverging phase are taken when the system size reaches at $N_{obs} = 20,000$. Clustering coefficient and nestedness

(marked by *) are given in those ratio to the ones of Erdős–Rényi random graph with the same size and average degree.

Reviewer's comment 3: In addition, there are some trivial comments for the improvement of the readability of the manuscript as follows:

-The authors mentioned their previous model (e.g., page 2, line 34, right), but it appears not always clearly explained in the manuscript. Could you clarify the original paper of that model and explain it briefly?

Author's response: Thank you for pointing out this. We agree that some parts mentioning the past works should be more specific. We added proper references and brief explanations.

(the middle of the first paragraph in Model, colored in red in the text)

... *The species with non-positive fitness, which in our **previous models [8,10]** went instantaneously extinct, will in the present model be inactivated after its fitness-dependent waiting time $\tau = e^f$, i.e. species in worse situation is inactivated faster.*

(the bottom of the first paragraph in Model, colored in red in the text)

... *Note that the present model with dormancy reduces to **the original model [8]** at $T_W = 0$.*

(the begging of the second paragraph in Model, colored in red in the text)

... *In **the previous models [8, 10]**, we added a new species every time the community has reached a persistent state.*

(the begging of the first paragraph in Result, colored in red in the text)

*Following the approach of **our previous study [8]**, we assess the robustness of the emergent system by the longterm trend of the system size, i.e. the number of species, under the successive introduction of new species **which, in terms of the directions and the weights of its interactions, has neutral effect on growth.** In our original model without any dormant mechanism [8], the system can grow limitlessly thus it is robust enough against the inclusion of new species, if the number of interactions given for each newly introduced species, m , is kept within a moderate range, i.e., $5 \leq m \leq 18$.*

(the middle of the second paragraph in Result, colored in red in the text)

... *Inheriting the nature of **the original model [8]**, the system with short dormancy limit T_W is found to be in the finite phase.*

(the begging of the fifth paragraph in Result, colored in red in the text)

***In the original model [8] and in the present model with $T_W = 0$** , in which the least fit species goes extinct first, the attacked resident species and the new species sequentially go extinct for $f - a < -b$ and otherwise only the new species goes extinct (i.e. is rejected).*

Reviewer's comment 4:

-The detail of FIG3 is not explained. Could you explain it in the main text or in its caption?

Author's response: Thank you for the suggestion. We added explanations for the detail of each panel.

(the caption of FIG3, colored in red in the text)

A limit cycle observed in an emergent system in the present model with inactivation and revival processes. (Top) The time series of the state changes of the species involved in the limit cycle (73, 80, and 90). (a) The entire system at the beginning of the cycle, $t = 0$. Red and blue arrows represent positive and negative interactions, respectively, such that the thickness represents the amplitude of the interaction. Precise values of the interactions are shown only for the interactions among the species 73, 80, and 90. (b) Species with negative fitness, 73, is inactivated at $t = \exp\{f_{73}\}$. This makes the fitness of species 80 and 90 non-positive. (c) Species with worse fitness, 90, is inactivated. This makes the fitness of species 73 positive. (d) Species 80 is later inactivated. (e) Species 73 is reactivated before the extinctions of species 80 and 90, making the fitness of those species positive. (f) Species 90 is reactivated. (g) Species 80 is reactivated and those three species come back to the initial all-active state.

Appendix B

Response to reviewer 2:

We would like to thank the referee for the review of our manuscript and the points of improvements. We are very grateful for your constructive comments, which are helpful in improving the manuscript. We have made revisions based on all these comments and suggestions. The followings are our point-by-point responses.

(Point-by-point response to the referee)

Reviewer: 2

Reviewer's comment 1: The authors aim to construct a model for studying the ecosystem of robustness with a small computational complexity. For this purpose, they introduced an inactive state of species with low fitness to the models in their previous studies. Inactive state is a feature in other ecosystem models which are computationally so costly. The authors claim that their extended model with a small computational complexity showed qualitatively similar behavior to these high-cost models.

Development of low-cost models is important for complex systems which require large-scale simulations.

My evaluation is that the paper is publishable with minor scientific revisions.

Author's response: Thank you for the careful read and the positive comment.

Reviewer's comment 2: My comments are as follows.

1. Introduction - 1st paragraph: The authors described their questions.

They should provide some references for the questions because these questions are general problems in statistical physics and complex system science.

Author's response: We have added some references for general point of view in this part, keeping other references which are based on more specific model in the original position.

(the begging of the first paragraph in Introduction, colored in red in the text)

The robustness of a system with many interacting elements or constituents under successive addition of new elements is an essential question for understanding the behaviour of various complex real world systems, that are often called ecosystems [1–3].

(the middle of the first paragraph in Introduction, colored in red in the text)

... This problem calls for a network theoretic approach, where the constituents of the system are the nodes of a dynamical network and the interactions are the links between them [4–7].

(references, colored in red in the text)

[2] Gardner MR, Ashby WR. 1970 Connectance of large dynamic (cybernetic) systems: critical values for stability. Nature 228, 784-784.

[3] Bak P, Sneppen K. 1993 Punctuated equilibrium and criticality in a simple model of evolution. *Phys. Rev. Lett.* **71**, 4083-4086.

[4] Pimm SL. 1984 The complexity and stability of ecosystems. *Nature* **307**, 321-326.

[5] Albert R, Jeong H, Barabási AL. 2000 Error and attack tolerance of complex networks. *Nature* **406**, 378-382.

[6] Schneider CM, Moreira AA, Andrade JS, Havlin S, Herrmann HJ. 2011 Mitigation of malicious attacks on networks. *Proc. Natl. Acad. Sci. USA* **108**, 3838-3841.

[7] Watanabe A, Mizutaka S, Yakubo K. 2015 *J. Phys. Soc. Jpn.* **84**, 114003.

Reviewer's comment 3:

2. Model: The authors extended their "original model".

However, they do not show a paper which introduced this model.

Which are papers [2-4] of the "original model"?

The authors should show the original paper and the differences between the original model and the extended model.

Author's response: Thank you for telling this point. We added proper references and brief explanations, which improved the readability.

(the middle of the first paragraph in Model, colored in red in the text)

... *The species with non-positive fitness, which in our **previous models** [8,10] went instantaneously extinct, will in the present model be inactivated after its fitness-dependent waiting time $\tau = e^f$, i.e. species in worse situation is inactivated faster.*

(the bottom of the first paragraph in Model, colored in red in the text)

... *Note that the present model with dormancy reduces to **the original model** [8] at $T_w = 0$.*

(the begging of the second paragraph in Model, colored in red in the text)

... *In **the previous models** [8, 10], we added a new species every time the community has reached a persistent state.*

(the third paragraph in Model, colored in red in the text)

Every time after finding a persistent state or elapsed time T_{int} , we proceed to the next time step by adding a new species with m interactions into the system. The m interacting species are chosen at random from the resident species with equal probability and the directions (incoming or outgoing) are also determined at random. The link weights are again assigned at random from the standard normal distribution.

Reviewer's comment 4:

3. Model: In the model, the parameter "a" showing relationships between species is important parameters. However, the description of "a" is missing.

Please add description how the model provides each "a" is not described in the main manuscript.

Author's response: We have recognized that the clear description of how to assigning " a_{ij} " for newly introduced species is only in Appendix and is missing in the main text. We have added the following sentences in the main text.

(the bottom of the third paragraph in Model, colored in red in the text)

... *The link weights are again assigned randomly from the standard normal distribution.*

Reviewer's comment 5:

4. Results: The authors sometimes mention their previous studies.

However, each mention is unclear which paper it refers to. For example, "our previous study" (L. 1 in 1st Paragraph) and "our original model without ..." (L. 4 in 1st Paragraph).

Author's response: We added proper references and brief explanations, together with your suggestion in the Model section.

(the begging of the first paragraph in Result, colored in red in the text)

*Following the approach of **our previous study [8]**, we assess the robustness of the emergent system by the longterm trend of the system size, i.e. the number of species, under the successive introduction of new species **which, in terms of the directions and the weights of its interactions, has neutral effect on growth.** In our original model without any dormant mechanism [8], the system can grow limitlessly thus it is robust enough against the inclusion of new species, if the number of interactions given for each newly introduced species, m , is kept within a moderate range, i.e., $5 \leq m \leq 18$.*

(the middle of the second paragraph in Result, colored in red in the text)

*... Inheriting the nature of **the original model [8]**, the system with short dormancy limit T_W is found to be in the finite phase.*

(the begging of the fifth paragraph in Result, colored in red in the text)

In the original model [8] and in the present model with $T_W = 0$, in which the least fit species goes extinct first, the attacked resident species and the new species sequentially go extinct for $f - a < -b$ and otherwise only the new species goes extinct (i.e. is rejected).

(the begging of the sixth paragraph in Result, colored in red in the text)

In the present model with $T_W > 0$, the situation is different as the resident species has another chance to reject such a falling-together attack.

Reviewer's comment 6: Here are the minor comments.

a. the footnote 1 and the reference 1 are duplicated.

Author's response: We have removed the footnote. Thank you.

Appendix C

Temporal inactivation enhances robustness in an evolving system

Fumiko Ogushi*

*Kyoto University Institute for Advanced Study, Kyoto University,
Yoshida Ushinomiya-cho, Sakyo-ku, Kyoto, 606-8501, JAPAN and
Center for Materials research by Information Integration,
National Institute for Materials Science, 1-2-1 Sengen, Tsukuba, Ibaraki 305-0047, JAPAN*

János Kertész†

*Department of Network and Data Science, Central European University, 1051 Budapest, Hungary and
Institute of Physics, Budapest University of Technology and Economics, 1111 Budapest, Hungary*

Kimmo Kaski‡

*Department of Computer Science, Aalto University School of Science, P.O. Box 15500, Espoo, Finland and
The Alan Turing Institute, British Library, 96 Euston Road, London NW1 2DB, UK*

Takashi Shimada§

*Mathematics and Informatics Center, The University of Tokyo and
Department of Systems Innovation, Graduate School of Engineering,
The University of Tokyo, 7-3-1 Hongo, Bunkyo-ku, Tokyo, 113-8656, JAPAN*

(Dated: January 12, 2019)

We study the robustness of an evolving system that is driven by successive inclusions of new elements or constituents with m random interactions to older ones. Each constitutive element in the model stays either active or is temporarily inactivated depending upon the influence of the other active elements. If the time spent by an element in the inactivated state reaches T_W , it gets extinct. The phase diagram of this dynamic model as a function of m and T_W is investigated by numerical and analytical methods and as a result both growing (robust) as well as non-growing (volatile) phases are identified. It is also found that larger time limit T_W enhances the system's robustness against the inclusion of new elements, mainly due to the system's increased ability to reject “falling-together” type attacks. Our results suggest that the ability of an element to survive in an unfavorable situation for a while, either as a minority or in a dormant state, could improve the robustness of the entire system.

Keywords: robustness, extinctions, network models, evolutionary dynamics, dormancy

I. INTRODUCTION

The robustness of a system with many interacting elements or constituents under successive addition of new elements is an essential question for understanding the behaviour of various complex real world systems, that are often called ecosystems [1–3]. In these systems the interactions between elements can be competitive or cooperative in nature such that the fitness of its elements or species can be strengthened or weakened by them, possibly causing the species getting extinct. This problem calls for a network theoretic approach, where the constituents of the system are the nodes of a dynamical network and the interactions are the links between them [4–7]. Then the rephrased question is about the evolution of such a network of nodes under the condition that new nodes with different kinds of links are introduced. If the network can grow, then the evolving system it describes

is considered robust, otherwise the system does not grow and is considered volatile. This way, we believe that the network approach can be used and be versatile in investigating various aspects of robustness for wide range of different systems.

Earlier it has been shown that in a simple model setting, where directed random positive and negative interactions characterize the system and the fitnesses of nodes (i.e. species) are identified with their strengths, when the links per node ratio —serving as a critical parameter— remains within a certain range, the system is robust [8]. This mechanism and the resulting phase diagram of the growth of the system were found to be universal, i.e., this feature is shared among a variety of models like the one with different distributions of interaction weights and with constant or random number of links introduced with the new nodes [9] and even with different bidirectional correlations [10]. While the range of robustness may be influenced by the details of the model, e.g., the mutual-ity in the interactions increasing it, the overall picture remains the same.

An alternative way to study the problem of robustness in complex interacting systems is population dynamics based approach as often done in theoretical ecology [11–

* ogushi.fumiko.54n@st.kyoto-u.ac.jp

† Kertesz.J@ceu.edu

‡ kimmo.kaski@aalto.fi

§ shimada@sys.t.u-tokyo.ac.jp

13]. Such a framework enables more complex dynamics and is flexible with respect to allowing different states of the species, but unlike in the network approach the inclusion of topological constraints are less straightforward in the population dynamics approach. Our aim here is to contribute to the convergence of these different approaches by including complex temporal features of interactions into the network models.

In population dynamics models, less fit species become minor in their population which in general makes that species almost irrelevant to the other species before that really gets extinct. For example, in the well adopted (generalized) Lotka-Volterra model [14, 15] and replicator dynamics model [16], the trajectory starting from a feasible initial state (i.e. all population variables are positive [17]) never touches 0 within finite time. Therefore, a threshold is generally introduced to model extinction. This is a simplified treatment of the Allee effect [18] about the weakening of the fitness in small populations, or rather direct modelling of the negative effect of demographic stochasticity [19, 20]. In summary, these observations and the related approaches suggest that the population size of less fit species and its temporal derivative becomes very small before extinction and the process is often lengthy. Furthermore, the adaptive nature of foraging and other interactions at the population level and at the individual level [13, 21–25] make such very minor species effectively even more invisible for other species. Therefore it seems plausible to include an “inactive state” into the set of possible states for handling such weakened populations. Species in such an inactive state, i.e. close to extinction, could be revived or reactivated within a frame of time if the circumstances would sufficiently improve.

The introduction of inactive state can be also regarded as modeling *dormancy*, which is broadly observed in biological ecosystems, such as in case of hibernation and surviving in seed, spore, or bacterial spore [26, 27]. From the evolutionary point of view hibernation or dormancy is favorable as it enables survival under scarce conditions.

FIG. 1. Introduction of the inactive state (dormancy) before the extinction, to our graph-dynamics framework. Less fit species is inactivated faster, and better fit species in inactive state is reactivated faster. The time limit of dormancy till extinction is, in contrast, uniformly set to T_W .

Therefore, we expect that this new component if considered in the framework of network models will increase the robustness of the system, which in turn should be reflected in the increase of the growth region in the phase diagram.

The paper is organized such that in the next section we describe our network based model of evolutionary system of species capable of being temporarily inactive. This is followed with a comprehensive account and analysis of computational modeling results to map out the phase diagram of the evolutionary system. Then we draw conclusion and present discussions.

II. MODEL

As we consider the ecosystems of being composed of connected species, we have devised our model being a network of nodes (or species) connected by unidirectional links with weights, as illustrated schematically in FIG. 1. Here the nodes represent species of animals of some sort and the links different types of directed influences between the pairs of species. The strength of the influence of species j on species i is denoted by the weight of the unidirectional link from node j to node i , i.e. a_{ij} . These weights can be either positive or negative. Each species has its “fitness”, which is simply given by the sum of its incoming interactions from other species in the system, i.e., $f_i = \sum_j^{\text{incoming}} a_{ij}$. A species can survive as long

as its fitness is greater than zero. The species with non-positive fitness, which in our previous models [8, 10] went instantaneously extinct, will in the present model be *inactivated* after its fitness-dependent waiting time $\tau = e^f$, i.e. species in worse situation is inactivated faster. The inactivated species loses its influence on other species thus we will neglect the links out of those for the calculation of fitness. If the surrounding community of an inactivated species changes and the fitness of an inactivated species becomes positive, the species is reactivated (waking up from dormancy). The waiting time of this reactivation process is also assumed to be fitness-dependent: $\tau = e^{-f}$. The slowest process among the microscopic dynamics is the inactivation and reactivation of solitary species ($f = 0$). The duration of these processes, $\tau = 1$, gives the unit of time to this otherwise timescale-less model. Although it is known that some species can maintain its dormancy for quite a long time [28], the period has generally a limit. In the following, we introduce a uniform time-limit parameter T_W . A species that has spent T_W of continuous time in the inactive state with non-positive fitness gets extinct. The extinct species and its incoming and outgoing links are removed permanently. Note that the present model with dormancy reduces to the original model [8] at $T_W = 0$. A pseudo-code style description of the entire dynamics is available in the Appendix.

FIG. 2. A temporal evolution of the model with inactivation (dormancy) and reactivation (revival), after inclusion of new species. (a): Introduction of a new species (red), which makes the fitness of two species (orange and magenta) negative. Each of these two species will be inactivated after its fitness-dependent duration: $\tau = e^{f_i}$. (b): Inactivation of the species with worse fitness (orange) takes place first and then the other species (magenta) is inactivated, which makes the fitness of another species (green) non-positive. Inactivated species is given T_w of waiting time till it will go extinct. (c): Green species is inactivated before any of other inactive species goes extinct. This change makes the fitness of the inactive species (magenta) positive. (d): Magenta species is reactivated after a fitness-dependent waiting time $\tau = e^{-f_i}$. Meanwhile, the orange species have spent T_w of time in the inactivated state and hence gone extinct: the orange species and the interactions from and to it are deleted. (e): Green species goes extinct. This does not change the sign of fitness of any species in the community. Therefore, after the extinction of green species, the system finally reaches to a new persistent state i.e. all the species are in the active state and have positive fitnesses. Nothing will happen for a community in a persistent state, until the next new species is introduced at $t + T_{\text{int}}$.

An example of temporal evolution of the system is shown in FIG 2. If all the species are in active state and have positive fitnesses, nothing will happen. Therefore we call such a state as a *persistent state*. In the previous models [8, 10], we added a new species every time the community has reached a persistent state. This corresponds to a low-introduction (mutation, invasion, etc) rate limit. In the present model, however, it is also possible that the system relaxes to a limit cycle and never reaches a *stationary* persistent state (FIG 3). Therefore,

FIG. 3. A limit cycle observed in an emergent system in the present model with inactivation and revival processes. (Top) The time series of the state changes of the species involved in the limit cycle (73, 80, and 90). (a) The entire system at the beginning of the cycle, $t = 0$. Red and blue arrows represent positive and negative interactions, respectively, such that the thickness represents the amplitude of the interaction. Precise values of the interactions are shown only for the interactions among the species 73, 80, and 90. (b) Species with negative fitness, 73, is inactivated at $t = \exp\{f_{73}\}$. This makes the fitness of species 80 and 90 non-positive. (c) Species with worse fitness, 90, is inactivated. This makes the fitness of species 73 positive. (d) Species 80 is later inactivated. (e) Species 73 is reactivated before the extinctions of species 80 and 90, making the fitness of those species positive. (f) Species 90 is reactivated. (g) Species 80 is reactivated and those three species come back to the initial all-active state.

we need a new parameter for the time interval of the species introduction, T_{int} . In the following, we take a long interval: $T_{\text{int}} = 100$ to keep a low-introduction rate, unless otherwise noted.

Every time after finding a persistent state or elapsed time T_{int} , we proceed to the next time step by adding

a new species with m interactions into the system. The m interacting species are chosen at random from the resident species with equal probability and the directions (incoming or outgoing) are also determined at random. The link weights are again assigned at random from the standard normal distribution.

III. RESULTS

Following the approach of our previous study [8], we assess the robustness of the emergent system by the long-term trend of the system size, i.e. the number of species, under the successive introduction of new species which, in terms of the directions and the weights of its interactions, has neutral effect on growth. In our original model without any dormant mechanism [8], the system can grow limitlessly thus it is robust enough against the inclusion of new species, if the number of interactions given for each newly introduced species, m , is kept within a moderate range, i.e., $5 \leq m \leq 18$. In contrast, the system with m outside this range, keeps fluctuating with a finite size. These fluctuations may lead to the extinction of the entire system and the lower the mean level is the higher is the probability for such an event. To avoid this possibility, we adopt an incubation rule when the system size becomes smaller than the initial system size N_0 . Under the incubation rule, we let totally isolated species (i.e. $f_i = 0$) stay in the active state or inactive state. This treatment prevents the total collapse of the system and provides the system with many more opportunities to search for growth from different initial conditions.

For sufficiently large initial system size, typically $N_0 \geq 100$, the limitless growth and finite size fluctuation behaviour are confirmed to be independent of the initial network structure. Therefore, we call the former behaviour taking place in the “diverging phase” and the latter in the “finite phase” of the parameter space. The temporal evolution of the system size of the present model with $m = 25$ is shown in FIG 4. Inheriting the nature of the original model [8], the system with short dormancy limit T_W is found to be in the finite phase. However, as T_W increases (to the value $T_W = 0.3$) the typical system size shows a clear increase yet it stays finite and for $T_W = 0.4$ and above the system has crossed a certain threshold to show diverging behaviour. This clearly illustrates that our newly introduced parameter T_W , the time limit for the continuous dormancy, can change the robustness of the system.

Next we will explore the whole phase diagram with systematic computer simulations by scanning through the m vs. T_W parameter space. The obtained phase diagram is shown in FIG 5, where it is seen that the introduction of dormancy and revival processes broaden the diverging phase. While this effect turns out to be larger for longer dormancy time limit T_W , yet it is not possible to get the system with very dense interactions ($m \geq 28$) to the diverging phase. It should also be noted that, as shown in

FIG. 4. The temporal evolutions of total number of active species $N_{\text{active}}(t)$ under the successive introduction of new species with $m = 25$ interactions. The unit for time is T_{int} i.e. the horizontal axis corresponds to the accumulated number of introduced species. The size of the emergent system diverges in time if the waiting time of dormancy is long ($T_W \geq 0.4$) while it fluctuates within a finite size for shorter waiting time ($T_W \leq 0.3$).

Appendix, the basic network characteristics of the emergent systems is not so much dependent on T_W and not deviated from that of Erdős-Rényi random graph, indicating the almost random network structure as in the previous models [8, 10].

The main mechanism of this enforcement is the rejection of “falling-together-attacks”. To illustrate this, let us consider a situation that a negative link weight ($-a$) is added to a resident species by a newly introduced species, which has zero or negative fitness value, $-b$ (FIG 6).

In the original model [8] and in the present model with $T_W = 0$, in which the least fit species goes extinct first, the attacked resident species and the new species sequentially go extinct for $f - a < -b$ and otherwise only the new species goes extinct (i.e. is rejected). Especially for the newly introduced species with no incoming links ($b = 0$, solitary attack), every attack strong enough ($f < a$) can kill the resident species before the newly introduced attacker species goes extinct.

In the present model with $T_W > 0$, the situation is different as the resident species has another chance to reject such a falling-together attack. The rejection happens if the resident species can survive in the inactivated state until the newly added species stays inactivated. The condition for this type of dynamics is as follows

$$f - a < -b < \ln(e^{f-a} + T_W). \quad (1)$$

Therefore, even a strong attack ($f > a$) by a solitary new species ($b = 0$) is rejected if $T_W > 1 - e^{f-a}$. And if $T_W \geq 1$, i. e. the limit of the dormancy period is long enough, even the solitary attacks never become successful. Note that the rejection acts perfectly in a special

FIG. 5. The phase diagram of the evolving open system with dormancy and revival processes. (Top): The speed of divergence $v = \lim_{t \rightarrow \infty} N(t)/t$ for the given original and new key parameters, m and T_W . The points where v is evaluated to be positive are shown by filled red symbols. (Bottom): The corresponding phase diagram.

case of $m = 1$, because in this situation every inclusion of new species corresponds to either a solitary attack or an attachment of species with no outgoing link. Therefore, even for this most sparse condition, large T_W drives the system with a mutually supporting community core to grow infinitely in size. However, such a growth is highly dependent on the initial condition (if there is no core in the initial network, the system collapses) which is out of the scope of this study. Thus we excluded this case from the phase diagram.

The increment of probability to reject falling-together-attacks directly contributes to the growth rate of the system, $v = N(t)/t$. A rough estimate of it near the upper phase boundary ($m \sim 18$) predicts a linear increase of the rejections to T_W for the small T_W regime (see Appendix for details), which is confirmed in the simulation (see FIG. 7). The observed contribution of the additional rejections to the system's growth rate, $\Delta v \sim T_W/8$, predicts the slope of the phase boundary to behave as $\Delta m^* \sim 20 T_W$. This is found to be consistent with the phase portrait.

FIG. 6. The mechanism of rejecting the attack by species with non-positive fitness.

FIG. 7. The rejection rate obtained from the simulation in the dense regime. In the small T_W regime shown here, the rejection rate increases linearly to T_W .

The effect of rejections in the sparse regime ($m \leq 4$) needs to be estimated differently. This is because the probability to have a solitary attack is larger. What is more significant, however, is the fact that the resident community has a sparse network structure, which in turn is very prone to a loss of certain species and can cause a cascade of extinctions of species supported by that species. Therefore, the effect of the increased chance of rejection can be more drastic. It is also possible that the structure of the emergent networks is changed, although **the basic network characteristics** (see Appendix)

FIG. 8. The cascade size distributions of the extinctions in the model with $m = 4$. The distributions from the systems in the finite phase ($T_W \leq 0.02$) and from the diverging phase ($T_W = 0.03$) overlap well each other, indicating the structure of the emerging networks is kept.

and the well kept distributions of extinction cascade size suggests it to be negligible at least for $m = 4$ (FIG. 8). The consideration above predicts the broadening of the diverging phase, but it is difficult to give an estimate of the effect of T_W against the very steep drop of the growth rate in this regime of the phase diagram.

IV. SUMMARY AND DISCUSSION

We have studied the robustness of an evolving system against successive inclusions of new elements or constituents, each with an ability to survive temporarily under unfavourable conditions in the state of being inactive. It is found that the introduction of the inactivation and revival processes broadens the phase the systems stays robust. This reinforcement of the emerging system is mainly due to its increased ability to reject falling-together type attacks. It should be noted that the broadening of the robust phase has a limit: systems with $m \geq 28$ stay in the finite phase even at $T_W = 1$, where the rejection probability reaches its maximum. The short term rejection process, in which a possible extinction of a species caused by the attack from a species with poor fitness is altered by the extinction of the attacker, can be regarded as a simplified dynamics in a class of population dynamics models [14–16, 23]. Because another type of interaction form, namely the ratio-dependent interaction [29], is known to reduce to our previous model [30], the extension of the model in this study has broadened the applicability of our theoretical framework. Similarly to our earlier results [8, 10], we have found that the number of interactions per species limits the system’s robustness. There are empirical findings in support to this ob-

servation [31].

As for the modelling in general the population dynamics models based on differential or difference state equations are able to describe rich evolutionary patterns following periodic and even chaotic trajectories, as observed in nature [32, 33]. However, this approach is generally computationally so costly that larger system sizes and longer time scales could not be studied. In order to circumvent these problems we have taken a network based approach, which is able to describe the dynamics of the system over much longer evolutionary time scale.

Although our present analysis covers up to the long-dormancy time limit ($T_W = 1$) in terms of the resulting short term rejection process, far longer dormancy limit ($T_W \gg T_{\text{int}}$) could bring new phenomena. Under such condition, inactive species can survive evolutionary time scale during which new species are introduced and that change the community. In some cases and for various kinds of systems, such as biological, social, and economic systems, it may be important to consider such long dormancy periods [36]. Also, the effect of bidirectionality [10] of the interaction should be examined, because it is expected to make the emergent system to show limit cycles more frequently. These two regimes, although that require heavier computation power, will reveal new phenomena and will better bridge with the continuous time dynamics models. Extending our approach so that some aspects of short term dynamics of more complex models is kept, with further spacial extension focusing on some aspects hardly accessible by traditional methods, is a promising way to treat evolutionary problems better [34, 35].

V. APPENDIX

A. Model procedure

- (0) (Create an initial system)
 - (i) Prepare N_0 species and connect them randomly by L_0 unidirectional links with link weights denoted by a_{ij} . Typical settings are $N_0 = 100$ and $L_0 = 10N_0$.
 - (ii) All species have its state variable ($S_i = \{-1, 1\}$, 1 and -1 denote active and inactive states, respectively), the time counters for state change g_i , and the counter for extinction h_i . Those are set to the initial values: $\{S_i\} = 1$, $\{g_i\} = 1$, $\{h_i\} = T_W$.
 - (iii) Set the system time at $t = 0$ and the time for the next new species introduction $T_{\text{next}} = T_{\text{int}}$.
- (1) Calculate the fitness f_i of each species,

$$f_i = \sum_j^{\text{incoming}} \left(\frac{1 + S_j}{2} \right) a_{ij}.$$

- (2) Reset the time counter if needed:

$$\begin{cases} g_i = 1 & (S_i = +1 \text{ and } f_i > 0 \cap f_i^{\text{old}} \leq 0) \\ g_i = 1 & (S_i = +1 \text{ and } f_i \leq 0 \cap f_i^{\text{old}} > 0) \\ g_i = 1 & (S_i = -1 \text{ and } f_i > 0 \cap f_i^{\text{old}} \leq 0) \\ h_i = T_W & (S_i = -1 \text{ and } f_i \leq 0 \cap f_i^{\text{old}} > 0) \end{cases}$$

where f_i^{old} is the fitness at the previous time step.

- (3) Calculate the remaining time till the next event for each species, δt_i :

$$\delta t_i = \begin{cases} T_{\text{next}} - t & (S_i = +1, f_i > 0 : \text{no state change}) \\ g_i e^{S_i f_i} & (S_i = +1, f_i \leq 0 : \text{inactivation}) \\ g_i e^{S_i f_i} & (S_i = -1, f_i > 0 : \text{reactivation}) \\ T_{\text{next}} - t & (S_i = -1, f_i \leq 0 : \text{extinction}). \end{cases}$$

- (4) Find the shortest time to the next event in the system: $\delta t_j^* = \min\{\delta t_i\}$.

- (5) Time translation of the system from t to $t + \delta t_j^*$

- (i) Update the system time $t = t + \delta t_j^*$
(ii) Update the time counters:

$$\begin{cases} g_i = g_i & (S_i = +1 \text{ and } f_i > 0) \\ g_i = g_i - e^{S_i f_i} \delta t_j^* & (S_i = +1 \text{ and } f_i \leq 0) \\ g_i = g_i - e^{S_i f_i} \delta t_j^* & (S_i = -1 \text{ and } f_i > 0) \\ h_i = h_i - \delta t_j^* & (S_i = -1 \text{ and } f_i \leq 0) \end{cases}$$

- (iii) Extinction: If $h_i \leq 0$, delete the species i and all links connecting to and from it.

- (6) Treat the event at t (state change of species j or new species introduction)

- If $t < T_{\text{next}}$, treat the nearest state change of species, j :
 - (i) Update the state of the species j :
 $S_j = -S_j$.
 - (ii) Reset the time counters:
 $g_j = 1$ and $h_j = T_W$.
- If $t = T_{\text{next}}$, add a new species:
 - (i) The new species is added in active state ($S = +1$) with the time counters $g = 1$ and $h = T_W$.
 - (ii) m interacting species are randomly chosen from the resident species.
 - (iii) The new species forms m directed unidirectional links. The direction of each new link is chosen with a equal probability 1/2.
 - (iv) The link weights are also randomly chosen from a standard normal distribution.
 - (v) Update the time for the next species introduction: $T_{\text{next}} = T_{\text{next}} + T_{\text{int}}$.

- (7) Recalculate the fitness: go back to step (1).

B. Estimation of the rate of the additional rejections and its effect

Here we first roughly estimate the increment of the chance to reject such falling-together-attack which directly contributes to the growth rate of the system, $v = N(t)/t$, near the upper phase boundary ($m \sim 18$). In the vicinity of the phase boundary in the dense regime, an inclusion of new species causes one strong attack ($f < a$) event in average. The distribution of $f - a$ is given by the negative side of the convolution:

$$\rho(f - a) = \int_0^\infty \bar{f}(\xi) G(1, f - a - \xi) d\xi, \quad (2)$$

where $\bar{f}(x)$ and $G(\sigma, x)$ represent the equilibrium fitness distribution of the emergent system and the Gaussian distribution with its standard deviation σ , respectively. The distribution of the fitness of newly added species, $-b$, is well approximated by the negative half side of the Gaussian distribution $G(\sqrt{m/2}, -b)$, where $m/2$ is the average number of incoming links. For small T_W , the condition to have the dormancy-aided rejection, Eq.(1), is

$$f - a < -b < f - a + \frac{T_W}{e^{f-a}}. \quad (3)$$

Substituting $\rho(-b)$ near 0 by its peak value $G(\sqrt{m/2}, 0) = 1/\sqrt{\pi m}$, and taking $f - a \sim -1/2$ as a typical attack strength, an estimated increment in the system's growth rate brought by the increase of the rejection is

$$\Delta v_{est.} \sim \sqrt{\frac{e}{\pi m}} T_W \sim \frac{T_W}{5}. \quad (4)$$

We can confirm this linear relation between the rejection rate Δv and T_W in the simulation results for $m = 19$ and 20 (FIG. 7). And the observed slope

$$\Delta v_{obs.} = \frac{T_W}{8} \quad (5)$$

is also consistent with the very rough estimation above.

Taking the linear slope of the system's intrinsic growth rate to m obtained from the observed growth rates,

$$\frac{\Delta v}{\Delta m} \sim \frac{0.06}{10}, \quad (6)$$

we reaches to an estimation for the slope of phase boundary

$$\Delta m \sim 20 T_W. \quad (7)$$

C. Network characteristics of the emergent systems

The network characteristics of emergent systems are shown in Table I.

m	T_W	$\langle N \rangle$	$\langle k \rangle$	assortativity	nestedness*	clustering coefficient*
19	0.5	$N_{obs} = 20000$	20.11 ± 0.05	-0.024 ± 0.001	1.0909 ± 0.0005	0.919 ± 0.005
	0.3	N_{obs}	20.04 ± 0.05	-0.0240 ± 0.0007	1.0884 ± 0.0003	0.928 ± 0.009
	0.1	N_{obs}	19.68 ± 0.05	-0.022 ± 0.001	1.0874 ± 0.0004	0.912 ± 0.007
	0.0	2.3×10^3	19.36 ± 0.01	-0.0197 ± 0.0001	1.0873 ± 0.0001	0.9172 ± 0.0001
10	0.5	N_{obs}	11.55 ± 0.03	-0.028 ± 0.001	1.1312 ± 0.0009	0.87 ± 0.02
	0.3	N_{obs}	11.39 ± 0.02	-0.030 ± 0.001	1.1288 ± 0.0007	0.89 ± 0.02
	0.1	N_{obs}	11.10 ± 0.02	-0.0338 ± 0.0007	1.1205 ± 0.0007	0.83 ± 0.02
	0.0	N_{obs}	10.89 ± 0.03	-0.036 ± 0.001	1.1188 ± 0.0005	0.86 ± 0.02
4	0.5	N_{obs}	5.20 ± 0.02	-0.040 ± 0.002	1.204 ± 0.002	0.75 ± 0.04
	0.3	N_{obs}	4.98 ± 0.01	-0.052 ± 0.001	1.186 ± 0.001	0.71 ± 0.09
	0.1	N_{obs}	4.69 ± 0.01	-0.073 ± 0.002	1.160 ± 0.001	0.72 ± 0.07
	0.0	1.0×10^3	4.63 ± 0.01	-0.0746 ± 0.0002	1.1567 ± 0.0003	0.662 ± 0.002
3	0.5	N_{obs}	4.04 ± 0.01	-0.058 ± 0.002	1.221 ± 0.002	0.59 ± 0.09
	0.3	N_{obs}	3.79 ± 0.01	-0.081 ± 0.001	1.187 ± 0.002	0.65 ± 0.05
	0.1	5.6×10^2	3.78 ± 0.01	-0.0748 ± 0.0009	1.1954 ± 0.0008	0.608 ± 0.006
	0.0	4.3×10^2	3.85 ± 0.01	-0.063 ± 0.002	1.204 ± 0.002	0.65 ± 0.01
2	0.5	N_{obs}	2.90 ± 0.01	-0.075 ± 0.003	1.256 ± 0.002	0.5 ± 0.1
	0.3	1.3×10^3	2.90 ± 0.01	-0.056 ± 0.001	1.265 ± 0.001	0.49 ± 0.02
	0.1	5.5×10^2	2.92 ± 0.01	-0.039 ± 0.004	1.271 ± 0.005	0.51 ± 0.02
	0.0	4.0×10^2	2.93 ± 0.03	-0.035 ± 0.008	1.296 ± 0.007	0.44 ± 0.06

TABLE I. Network characteristics of emergent systems. For each parameter set, (m, T_W) , the average and the error are calculated from 10 independent simulation samples. Samples in the diverging phase are taken when the system size reaches at $N_{obs} = 20,000$. The clustering coefficient and nestedness (marked by *) are given in those ratio to the ones of Erdős-Rényi random graph with the same size and average degree.

ETHICS

This study did not require ethical approval.

DATA ACCESSIBILITY

The simulation code for our model has been uploaded as the electronic supplemental material.

AUTHOR'S CONTRIBUTIONS

F.O. and T.S. conceived the model and conducted the simulation. All authors analysed the results and wrote the manuscript.

COMPETING INTERESTS

We declare we have no competing interests.

FUNDING

F.O. was partly supported by "Materials Research by Information Integration" Initiative (MI2I) project of

the Support Program for Starting Up Innovation Hub from the Japan Science and Technology Agency (JST). K.K. acknowledges financial support by the Academy of Finland Research project (COSDYN) No. 276439, EU HORIZON 2020 FET Open RIA project (IBSEN) No. 662725, EU HORIZON 2020 INFRAIA-1-2014-2015 program project (SoBigData) No. 654024, and the Rutherford Foundation Visiting Fellowship at The Alan Turing Institute, UK. T.S. was partly supported by JSPS KAKENHI Grant Number 15K05202 and 18K03449.

ACKNOWLEDGMENTS

FO, JK, and TS thank for hospitality of Aalto University.

DISCLAIMER

Any opinions, findings or conclusions are those of authors.

[1] Here the term "ecosystem" is used in a rather general sense to mean biological ecosystems but also diverse eco-

nomical and social systems of individuals and institu-

- tions.
- [2] Gardner MR, Ashby WR. 1970 Connectance of large dynamic (cybernetic) systems: critical values for stability. *Nature* **228**, 784-784.
 - [3] Bak P, Sneppen K. 1993 Punctuated equilibrium and criticality in a simple model of evolution. *Phys. Rev. Lett.* **71**, 4083-4086.
 - [4] Pimm SL. 1984 The complexity and stability of ecosystems. *Nature* **307**, 321-326.
 - [5] Albert R, Jeong H, Barabási AL. 2000 Error and attack tolerance of complex networks. *Nature* **406**, 378-382.
 - [6] Schneider CM, Moreira AA, Andrade JS, Havlin S, Herrmann HJ. 2011 Mitigation of malicious attacks on networks. *Proc. Natl. Acad. Sci. USA* **108**, 3838-3841.
 - [7] Watanabe A, Mizutaka S, Yakubo K. 2015 *J. Phys. Soc. Jpn.* **84**, 114003.
 - [8] Shimada T. 2014 A universal transition in the robustness of evolving open systems. *Scientific Reports* **4**, 4082. (doi:10.1038/srep04082)
 - [9] Shimada T. 2015 *A Universal Mechanism of Determining the Robustness of Evolving Systems*. In *Mathematical Approaches to Biological Systems*, pp. 95-117. Springer Japan. (doi:10.1007/978-4-431-55444-8_5)
 - [10] Ogushi F, Kertész J, Kaski K, Shimada T. 2017 Enhanced robustness of evolving open systems by the bidirectionality of interactions between elements. *Scientific Reports* **7**, 6978. (doi:10.1038/s41598-017-07283-9)
 - [11] Maynard Smith J. 1982 *Evolution and the theory of games*. Cambridge, UK: Cambridge University Press
 - [12] Murray JD. 2001 *Mathematical Biology: I. An Introduction*, third edition ed. Berlin, Germany: Springer.
 - [13] Begon M, Townsend CR, Harper JL. 2005 *Ecology: From Individuals to Ecosystems*. 4th edn. Oxford, UK: Blackwell Publishing.
 - [14] Taylor PJ. 1988 Consistent scaling and parameter choice for linear and Generalized Lotka-Volterra models used in community ecology. *Journal of Theoretical Biology* **135**, 543-568. (doi:10.1016/s0022-5193(88)80275-3)
 - [15] Taylor PJ. 1988 The construction and turnover of complex community models having Generalized Lotka-Volterra dynamics. *Journal of Theoretical Biology* **135**, 569-588. (doi:10.1016/s0022-5193(88)80276-5)
 - [16] Tokita K, Yasutomi A. 1999 Mass extinction in a dynamical system of evolution with variable dimension. *Physical Review E* **60**, 842-847. (doi:10.1103/physreve.60.842)
 - [17] Roberts A. 1974 The stability of a feasible random ecosystem. *Nature* **251**, 607-608. (doi:10.1038/251607a0)
 - [18] Stephens PA, Sutherland WJ, Freckleton RP. 1999 What Is the Allee Effect? *Oikos* **87**, 185. (doi:10.2307/3547011)
 - [19] Melbourne BA, Hastings A. 2008 Extinction risk depends strongly on factors contributing to stochasticity. *Nature* **454**, 100-103 (doi:10.1038/nature06922)
 - [20] Murase Y, Shimada T, Ito N, Rikvold PA. 2010 Effects of demographic stochasticity on biological community assembly on evolutionary time scales. *Physical Review E* **81**, 041908. (doi:10.1103/physreve.81.041908)
 - [21] Hori M. 1993 Frequency-Dependent Natural Selection in the Handedness of Scale-Eating Cichlid Fish. *Science* **260**, 216-219 (doi:10.1126/science.260.5105.216)
 - [22] Dukas R, Bernays EA. 2000 Learning improves growth rate in grasshoppers. *Proceedings of the National Academy of Sciences* **97**, 2637-2640. (doi:10.1073/pnas.050461497)
 - [23] Kondoh M. 2003 Foraging adaptation and the relationship between food-Web Complexity and stability. *Science* **299**, 1388-1391 (doi:10.1126/science.1079154)
 - [24] Takeuchi Y, Oda Y. 2017 Lateralized scale-eating behaviour of cichlid is acquired by learning to use the naturally stronger side. *Scientific Reports* **7**, 8984. (doi:10.1038/s41598-017-09342-7)
 - [25] Stuart YE, Campbell TS, Hohenlohe PA, Reynolds RG, Revell LJ, Losos JB. 2014 Rapid evolution of a native species following invasion by a congener. *Science* **346**, 463-466. (doi:10.1126/science.1257008)
 - [26] Andrews MT. 2007 Advances in molecular biology of hibernation in mammals. *BioEssays* **29**, 431-440. (doi:10.1002/bies.20560)
 - [27] Maps F, Runge JA, Leising A, Pershing AJ, Record NR, Plourde S, Pierson JJ. 2011 Modelling the timing and duration of dormancy in populations of *Calanus finmarchicus* from the Northwest Atlantic shelf. *Journal of Plankton Research* **34**, 36-54. (doi:10.1093/plankt/fbr088)
 - [28] Tsujimoto M, Imura S, Kanda H. 2016 Recovery and reproduction of an Antarctic tardigrade retrieved from a moss sample frozen for over 30 years. *Cryobiology* **72**, 78-81. (doi:10.1016/j.cryobiol.2015.12.003)
 - [29] Arditi R, Ginzburg LR. 1989 Coupling in predator-prey dynamics: Ratio-Dependence. *Journal of Theoretical Biology* **139**, 311-326. (doi:10.1016/s0022-5193(89)80211-5)
 - [30] Shimada T, Murase Y, Ito N. 2015 Do Connections Make Systems Robust? A New Scenario for the Complexity-Stability Relation. In *Proceedings of the International Conference on Social Modeling and Simulation, plus Econophysics Colloquium 2014*, pp. 99-109. Springer International Publishing. (doi:10.1007/978-3-319-20591-5_9)
 - [31] Ings TC et al. 2009 Review: Ecological networks - beyond food webs. *Journal of Animal Ecology* **78**, 253-269. (doi:10.1111/j.1365-2656.2008.01460.x)
 - [32] Becks L, Hilker FM, Malchow H, Jürgens K, Arndt H. 2005 Experimental demonstration of chaos in a microbial food web. *Nature* **435**, 1226-1229. (doi:10.1038/nature03627)
 - [33] Benincà E, Huisman J, Heerkloss R, Jöhnk KD, Branco P, Van Nes EH, Scheffer M, Ellner SP. 2008 Chaos in a long-term experiment with a plankton community. *Nature* **451**, 822-825. (doi:10.1038/nature06512)
 - [34] Kuussaari M et al. 2009 Extinction debt: a challenge for biodiversity conservation. *Trends in Ecology & Evolution* **24**, 564-571. (doi:10.1016/j.tree.2009.04.011)
 - [35] Gonzalez A, Bell G. 2012 Evolutionary rescue and adaptation to abrupt environmental change depends upon the history of stress. *Philosophical Transactions of the Royal Society B: Biological Sciences* **368**, 20120079-20120079. (doi:10.1098/rstb.2012.0079)
 - [36] Vreeland R, Straight S, Krammes J, Dougherty K, Rosenzweig W, Kamekura M. 2002 Halosimplex carlsbadense gen. nov., sp. nov., a unique halophilic archaeon, with three 16S rRNA genes, that grows only in defined medium with glycerol and acetate or pyruvate. *Extremophiles* **6**, 445-452. (doi:10.1007/s00792-002-0278-3)